# YOLO-RD: Introducing Relevant and Compact Explicit Knowledge to YOLO by Retriever-Dictionary

**Hao-Tang Tsui, Chien-Yao Wang & Hong-Yuan Mark Liao**
Institute of Information Science, Academia Sinica
{henrytsui,kinyiu,liao}@iis.sinica.edu.tw

## Abstract

Identifying and localizing objects within images is a fundamental challenge, and numerous efforts have been made to enhance model accuracy by experimenting with diverse architectures and refining training strategies. Nevertheless, a prevalent limitation in existing models is overemphasizing the current input while ignoring the information from the entire dataset. We introduce an innovative ***Retriever-Dictionary*** (RD) module to address this issue. This architecture enables YOLO-based models to efficiently retrieve features from a Dictionary that contains the insight of the dataset, which is built by the knowledge from Visual Models (VM), Large Language Models (LLM), or Visual Language Models (VLM). The flexible RD enables the model to incorporate such explicit knowledge that enhances the ability to benefit multiple tasks, specifically, segmentation, detection, and classification, from pixel to image level. The experiments show that using the RD significantly improves model performance, achieving more than a 3% increase in mean Average Precision for object detection with less than a 1% increase in model parameters. Beyond YOLO, the RD module improves the effectiveness of 2-stage models and DETR-based architectures, such as Faster R-CNN and Deformable DETR. Code is released at `https://github.com/henrytsui000/YOLO`.

## 1 Introduction

In the field of computer vision, object detection models play a pivotal role, these models are designed to precisely locate objects within images. They are used in applications such as medical image analysis and autonomous driving. Additionally, they can also be used as backbone models for downstream tasks like multi-object tracking (Zhang et al., 2022; Cao et al., 2023; Aharon et al., 2022), and crowd counting (Zhang et al., 2016; Song et al., 2021). As the basis for these extended tasks, object detection models must combine high accuracy with low latency to allow downstream tasks to stand on the shoulders of giants.

Among object detection models, the YOLO (Redmon et al., 2016), FasterRCNN (Ren et al., 2015), and DETR (Carion et al., 2020) are notably prevalent. The YOLO series primarily utilizes Convolutional Neural Networks (CNN) (LeCun et al., 1998), providing a balance between inference speed and accuracy. From YOLOv1 through YOLOv10 (Redmon et al., 2016; Redmon & Farhadi, 2017; 2018; Bochkovskiy et al., 2020; Jocher, 2020; Li et al., 2022; Wang et al., 2023a; Jocher et al., 2023; Wang et al., 2024b;a), there has been a consistent focus on refining the architecture and training methods to reduce the model's parameters while enhancing accuracy. Beyond the main YOLO series, variants such as YOLOR (Wang et al., 2023b) incorporate implicit knowledge and other techniques to further boost model performance.

As shown in Figure 1, while both CNNs and Transformers (Vaswani et al., 2017) concentrate on the input image, CNNs are restricted to local input data, and Transformers, despite considering interactions among various inputs, are still confined to the given inputs or other model branches. However, the above-mentioned models often overlook a crucial aspect of explicit knowledge—the comprehensive dataset information. On the other hand, some contrastive methods, e.g. SimCLR (Chen et al., 2020), DINO (Caron et al., 2021) have demonstrated that cross-referencing data is beneficial.

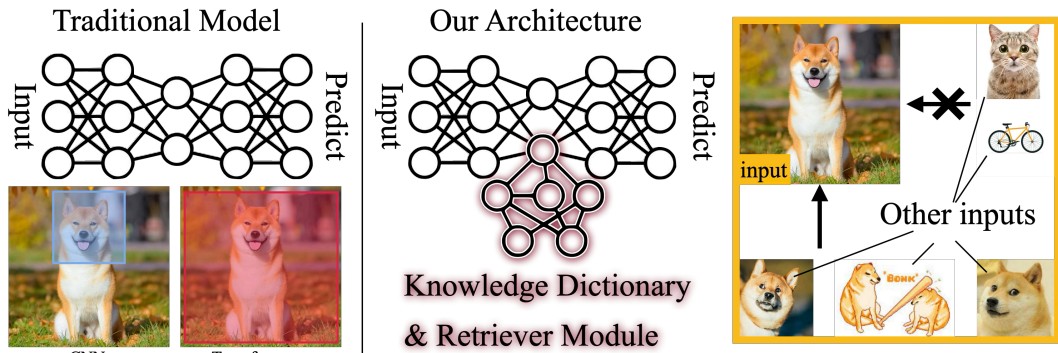

Figure 1: A comparison between traditional models and our proposed **Retriever Dictionary** module. On the left, CNN-based models focus on local regions (blue box), while Transformer-based models tend to utilize the entire image (red box). However, both methods fail to leverage the information from the entire dataset, as illustrated in the bottom-right corner. Our module enhances the model's resource utilization by incorporating knowledge from other parts of the dataset.

Moreover, in the realm of natural language processing, some models incorporate the Retrieval Augmented Generation (RAG) architecture (Lewis et al., 2020), which stores knowledge in a pre-built database and retrieves this information during inference to pass to the generator for encoding. This allows the model to access external information from a large, pre-established dataset. This approach significantly enhances the capabilities of large language models, although it typically requires substantial computational resources. However, applying this technique to object detection or other computer vision tasks still faces significant challenges, particularly in preparing and retrieving external data. Object detection models must carefully balance accuracy, model parameters, and latency when handling external information.

To address these challenges, we introduce a compact external module composed of a **Retriever** and a **Dictionary**, designed to enhance dataset utilization in computer vision models during training. This module effectively filters out irrelevant information and amplifies crucial data. The **Retriever** aggregates region features to generate a query, while the **Dictionary**, containing comprehensive dataset information, enables the query to select relevant atoms. Notably, this pre-built **Dictionary** extends beyond the YOLO backbone, incorporating data encoders like VLMs or LLMs, which bring extensive training data and knowledge for more precise and comprehensive information.

This module allows models to reinforce data during the forward process, benefiting not only region-level tasks like object detection but also pixel-level tasks like segmentation and whole-image tasks like classification. Furthermore, our module can be extended to various model architectures, such as the FPN network in Faster RCNN and the backbone-encoder regions in Detection Transformers, providing higher-quality information during downsampling, ultimately leading to much better performance.

In this paper, we make several key contributions:

- We introduce a **Retriever-Dictionary** module that enables efficient utilization of external information without the need for an external loss function, while still maintaining the **Dictionary**'s properties and allowing updates to its parameters.

- We demonstrate that incorporating external knowledge from models such as VLMs and LLMs can significantly enhance model performance.

- We further show that integrating external information improves not only YOLO's object detection performance but also other mainstream vision tasks and architectures.

These improvements substantially enhance the capabilities of object detection models and demonstrate that our module exhibits All-to-All properties, allowing it to utilize external knowledge to improve performance across multiple tasks and model architectures with minimal parameters.

## 2 RELATED WORK

**Real-time object detection.** Real-time object detection is a foundational problem in computer vision, with a focus on achieving low latency and high accuracy. Traditional approaches have primarily focused on CNN architectures, with seminal works such as the OverFeat (Sermanet et al., 2014) and Faster R-CNN(Girshick, 2015; Ren et al., 2015). Since the introduction of Vision Transformers (ViT) (Dosovitskiy et al., 2021), there have been notable follow-up works, including DETR (Carion et al., 2020) and RT-DETR (Zhao et al., 2024). However, the CNN-based YOLO series models hold a crucial position in the field of real-time detection due to their ease of training from scratch, lightweight design, and ability to perform high-speed inference.

Each version of the YOLO model introduces different architectures and training strategies. For example, YOLOv7 (Wang et al., 2023a) employs ELAN and trainable bag-of-freebies techniques to enhance performance, while YOLOv9 (Wang et al., 2024b) incorporates Generalized Efficient Layer Aggregation Networks (G-ELAN) and Programmable Gradient Information for improved efficiency and learning capability. YOLOv10 (Wang et al., 2024a) further introduces the compact inverted block to optimize model size and computation. On the theoretical front, works like YOLOR (Wang et al., 2023b) leverage the shared characteristics across multiple computer vision tasks, allowing the model to learn implicit knowledge and relax the prediction head, thus generalizing the YOLO architecture to various tasks. Despite these architectural and strategic differences, all YOLO models share a conceptual framework comprising three core modules: a Backbone for downsampling, a Neck (e.g. FPN) for feature fusion, and a Detection Head for final prediction. In this paper, we also utilize the Backbone as an image encoder.

**Dictionary learning.** Dictionary learning is a fundamental technique in signal processing and machine learning, aimed at learning a set of basis functions (or atoms) that can efficiently represent signals. This technique has been extensively explored in the context of sparse coding, where signals are approximated as sparse linear combinations of dictionary atoms. Additionally, it was discovered that natural images can be effectively represented using sparse coding models (Olshausen & Field, 1997), laying the foundation for further development of dictionary learning algorithms.

Several algorithms have been proposed to optimize dictionaries and sparse coefficients. Among these, K-SVD (Aharon et al., 2006) has become a standard due to its effectiveness in applications such as image denoising, compression, and inpainting. With the rise of CNNs, dictionary learning has also seen new developments, such as designing convolutional blocks and defining loss functions to achieve dictionary learning objectives (Garcia-Cardona & Wohlberg, 2018; Zheng et al., 2021). Additionally, dictionary learning has been applied to tasks like content-based image retrieval (CBIR), as seen in works like Şaban Öztürk (2021); Tarawneh et al. (2019). In this paper, we focus on dictionary learning rather than sparse dictionary learning and emphasize dictionaries that can robustly represent information and retain critical, relevant signals.

**Retrieval-augmented generation (RAG)** Retrieval-Augmented Generation (RAG) (Lewis et al., 2020) is a technique that first appeared in large language models. It primarily involves three steps: Indexing, where the database is split into chunks, encoded into vectors, and stored in a vector database; Retrieval, which retrieves relevant information based on similarity to the input; and Generation, where both the original input and the retrieved information are fed into the model for further processing. This approach enables LLM to handle unseen information and has been successfully applied in T5 (Raffel et al., 2020), or certain versions of ChatGPT(Achiam et al., 2023).

Due to the time-consuming nature of the retrieval process, and the real-time requirements of most computer vision tasks, RAG has seen limited application in vision-based models. Recently, some work has been done to mitigate the bottleneck in Wu & Xie (2024); Kim et al. (2024); Liu et al. (2023). For example, RALF (Kim et al., 2024) leverages LLM to embed a huge vocabulary set and find similar meaning words to refine features; REACT (Liu et al., 2023) utilizes the World Wide Web as the information source to extend model knowledge. However, they still require an undetachable huge dataset or Language model which leads to extremely high training and inference loading. In this work, we provide the vision-based model with not only a light-weight database but also refinable features.

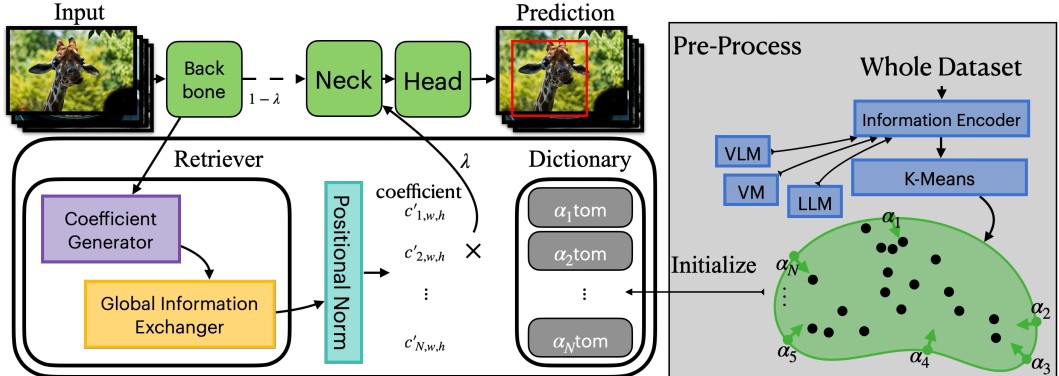

Figure 2: **Dictionary** is initialized by encoding the dataset using an image encoder, from which $N$ embeddings are selected as **Dictionary** atoms. During training, for each input feature $X_{w,h}$, the **Retriever** core—comprising the Coefficient Generator (**G**) and the Global Information Exchanger (**E**)—generates coefficients for each atom $\alpha$ in the **Dictionary** $D$. Then, the normalized coefficients are used as weights for each **Dictionary** atom. Finally, by concatenating the residual of $X_{w,h}$, the output $Y_{w,h}$ is obtained.

## 3 METHOD

In this work, we introduced the **Retriever-Dictionary** module, as shown in Figure 2, which enables computer vision models to utilize comprehensive dataset knowledge with minimal extra parameters quickly. This plug-in stores encoded information from various models, enhancing the model's ability to identify which features of the input data should be emphasized or diminished, thereby improving overall performance. The RD-module is composed of two main components: the **Retriever** and the **Dictionary**. The **Dictionary** consists of $N$ elements, each represented as $\mathbb{R}^f$ vectors, known as atoms $\alpha$. The **Retriever** generates the coefficient of $\alpha$ for each pixel according to the input. The **Dictionary** can be generated using the YOLO backbone or initialized with diverse models like the visual language model CLIP (Radford et al., 2021) and large language models such as GPT (Radford et al., 2018). This approach allows the model to align visual and linguistic representations, leading to a more balanced and valuable distribution of atoms. Furthermore, the incorporation of linguistic knowledge helps the module retain crucial information. The main goal is to adjust the distribution of the **Dictionary** and allow the **Retriever** core to find the best Atoms' weight for each input pixel.

### 3.1 MODULE STRUCTURE

The **Retriever** core aims to efficiently generate the coefficients of each $\alpha$ in the **Dictionary**. Inspired by depthwise convolution Chollet (2017), we separate the **Retriever** into two components: the Coefficient Generator **G** and the Global Information Exchanger **E**. The Coefficient Generator, denoted as $\mathbf{G} : \mathbb{R}^{f \times W \times H} \to \mathbb{R}^{N \times W \times H}$, computes coarse coefficients based on the input feature map $X \in \mathbb{R}^{f \times W \times H}$, where $f$ is the input feature dimension and $W$, $H$ represent the width and height of the feature map, respectively. The coarse coefficients are calculated as follows:

$$Y = \mathbf{G}(X) = W^G \cdot X_{w,h}, \tag{1}$$

where $W^G \in \mathbb{R}^{N \times f}$ is the projection matrix of **G**, and $X_{w,h} \in \mathbb{R}^f$ is the feature vector at spatial location $(w, h)$.

The Global Information Exchanger, denoted as $\mathbf{E} : \mathbb{R}^{N \times W \times H} \to \mathbb{R}^{N \times W \times H}$, refines and exchanges information across neighboring pixels, and is defined as:

$$\mathbf{E}(Y) = W^{E^{(i)}} * Y^{(i)}, \tag{2}$$

where $W^E \in \mathbb{R}^{N \times 1 \times k \times k}$ is the depthwise convolution filter with kernel size $k \times k$, and $i \in [0, N)$ indexes the channels of $Y$. The term $Y^{(i)} \in \mathbb{R}^{W \times H}$ refers to the spatial feature map of the $i$-th channel.

The combined operation of $\mathbf{G}$ and $\mathbf{E}$ generates the final coefficients vector $c$ for each pixel:

$$c = \mathbf{E}(\mathbf{G}(X_{w,h})). \tag{3}$$

This separation of tasks minimizes direct computation of the coefficient vectors $c$ at each pixel location, significantly reducing the parameter count while maintaining high performance (see Experiments 4.3 and Appendix A.8 for further discussion).

To prevent the coefficient vectors $c$ from simply replicating the input features, which would make the *Dictionary* become an identity matrix, we normalize each $c$. We apply a normalization process equivalent to Positional Normalization (PONO) (Li et al., 2019), which is defined as:

$$\text{PONO}(X) = \frac{X - \mu_c}{\sqrt{\sigma_c + \epsilon}} \cdot \gamma + \beta, \tag{4}$$

where the mean $\mu_c$ and variance $\sigma_c$ are calculated as:

$$\mu_c = \frac{1}{N} \sum_{n=1}^{N} X_n, \quad \sigma_c = \frac{1}{N} \sum_{n=1}^{N} (X_n - \mu_c)^2,$$

with $X_n$ representing the $n$-th feature at a fixed spatial location, and $\gamma$, $\beta$ are learned parameters. Although PONO was initially designed to preserve structural information in generative networks, here it ensures that the feature vectors $c$ are properly scaled and centered, preventing the *Dictionary* from collapsing into an identity matrix.

The normalized coefficients $c'$ are then used to select atoms from the *Dictionary*, either enhancing or diminishing specific features. This selection is a weighted summation of the atoms and integration with the input residuals to produce the final output. To preserve the dictionary's learning dynamics, each atom is normalized to unit length during training. The resulting formula is as follows:

$$Y_{h,w} = \lambda \cdot X_{h,w} + (1 - \lambda) \cdot \sum_{i=1}^{N} \mathbf{c}'_{i,h,w} \cdot \alpha_i, \tag{5}$$

where $|\alpha_i| = 1$ for all $\alpha_i \in D$, and $\lambda$ is the residual weight. As can be seen in Equation 5, the weighted summation of *Dictionary* atoms is mathematically equivalent to a convolutional layer with a kernel size of 1, stride of 1, and no bias term. To ensure that the sum of each atom's components equals 1, as required in dictionary learning, we employ weight normalization (Salimans & Kingma, 2016). This eliminates the need for an external objective function to enforce this condition. Weight normalization is defined as:

$$\text{WN}(D) = \left\{ \frac{\alpha}{|\alpha|} \mid \forall \alpha \in D \right\}. \tag{6}$$

Finally, we combine the *Retriever* core ($\mathbf{G}$ and $\mathbf{E}$), the *Dictionary*, and the residual connection to express the entire *Retriever Dictionary* process $\text{RD} : \mathbb{R}^{f \times W \times H} \to \mathbb{R}^{f \times W \times H}$ as:

$$Z = \text{RD}(X) = \lambda \cdot X + (1 - \lambda) \cdot \text{PONO}(\mathbf{E}(\mathbf{G}(X))) * \text{WN}(D), \tag{7}$$

where $*\text{WN}(D)$ denotes the convolution operation using the weight-normalized *Dictionary* atoms set $\text{WN}(D)$ as filters.

## 3.2 *Dictionary* INITIALIZATION

By pre-initializing the *Dictionary*, we embed knowledge into the module atoms of the *Dictionary*. More precisely, we use the selected encoder to map the entire dataset into a high-dimensional space, by incorporating insights from various modality models, as illustrated in Figure 3. This high-dimensional space shares the same dimension as the original YOLO backbone middle layer. Obviously, this will have a huge number of vectors and present multiple groups in high-dimensional space, we employ $k$-means (Macqueen, 1967) to leave representative vectors, which ultimately serve as *Dictionary* atoms. Through these operations, we can map the dataset pairs into a high-dimensional space in a short time (this operation is equivalent to using the model to traverse the entire dataset at the speed of inference), and find vectors that can represent most of the dataset. For features that are not in the *Dictionary*, we can use the feature through the residual mechanism in the module, and utilize the *Retriever Dictionary* to bring closer atoms of the same category to the outlier feature. Next, we discuss using different modality models as encoders for encoding knowledge:

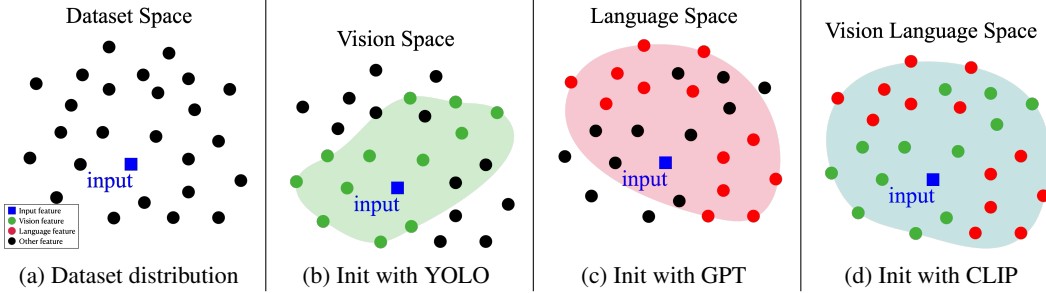

Figure 3: Illustrates the distribution of the dataset in the model's middle layer, where the blue square represents the current input feature. In traditional models, only the input feature is used, neglecting the rich information available in the dataset. In contrast, with our Retriever-Dictionary model, additional data information is retrieved from the dataset. The dictionary can be initialized from different models: vision models, language models, or vision-language models. The latter provides a more comprehensive and integrated representation of the dataset.

**Vision model.**    In the YOLO architecture, the backbone layer is designed to transform the original image into high-dimensional embeddings, providing the FPN with enriched regional information, which makes the backbone an ideal image encoder. Therefore, we use the modern pre-trained YOLOv9 (Wang et al., 2024b) as the vision model encoder. This model is leveraged to traverse the entire training set, converting all the dataset's data into feature-dimensional distributions.

**Vision language model.**    As a visual language model, CLIP possesses a strong understanding of input image features and can map these features into a semantically related space for comparison. Therefore, we utilize CLIP for visual language initialization. However, according to Dense-CLIP (Rao et al., 2022), although CLIP's output embeddings are global representations of the image, the output of CLIP's ViT blocks retains information about corresponding regions. CLIP primarily uses the first patches as embeddings, leaving out local information from the other patches. As a result, we leverage all the output patches from CLIP's image encoder to embed the dataset more comprehensively.

**Large language model.**    For initialization from an LLM, since language models cannot directly convert images into feature embeddings and image captions typically describe the entire image rather than specific regions, we use class names from MSCOCO (Lin et al., 2014) and ImageNet-21k (Ridnik et al., 2021), along with image captions from MSCOCO, as prompts for GPT (Radford et al., 2018). However, because some feature dimensions deviate significantly from the unit interval, we apply standard normalization to scale each dimension, aiming to stabilize the training process while preserving the relative positioning of the vectors.

### 3.3   *Dictionary* COMPRESSION

The objective of designing *Retriever Dictionary* (RD) is to retain the most critical information from the dataset. However, even after training, the number of atoms in the *Dictionary* may exceed the required amount, with some atoms being infrequently used or irrelevant to the specific dataset domain. Drawing inspiration from knowledge distillation and the Teacher-Student model (Hinton et al., 2015), we condense the original *Dictionary* $D$ into a smaller and more efficient version, denoted as $d$. To align the output features of $d$ with those of $D$, we employ contrastive learning (Chen et al., 2020) to provide the $d$ soft labels instead of the traditional cross-entropy loss. In this process, the model backbone and RD are frozen, while the optimization is focused on the smaller *Retriever Dictionary* module $d$. The objective function is defined as:

$$\mathcal{L}_{i,w,h} = -\log \frac{\exp(\text{sim}(z_{i,w,h}^{\text{rd}}, z_{i,w,h}^{\text{RD}})/\tau)}{\sum_{j=1}^{B} \sum_{w',h'}^{W,H} \exp(\text{sim}(z_{i,w,h}^{\text{rd}}, z_{j,w',h'}^{\text{RD}})/\tau)}, \tag{8}$$

Table 1: Comparing the performance and the improvements across different modalities of RD-module and structures on the COCO 2017 validation set.

| BackBone | Initializer | Params | Latency | $f$ | mAP$_{.5:.95}^{val}$ (%) | mAP$_{.5}^{val}$(%) |
|---|---|---|---|---|---|---|
| YOLOv7 | Origin | 37.2M | 3.59 | - | 50.04 | 68.65 |
| YOLOv9 | Origin | 25.3M | 4.00 | - | 52.64 | 69.56 |
| Faster-RCNN | Origin | 43.1M | 41.00 | - | 38.40 | 59.00 |
| Deformable DETR | Origin | 40.1M | 41.10 | - | 43.80 | 62.60 |
| YOLOv7 | VM | 37.4M | 3.70 | $\mathbb{R}^{512}$ | 51.37 (↑ 2.66%) | 69.42 (↑ 1.13%) |
| YOLOv9 | VM | 25.5M | 4.16 | $\mathbb{R}^{512}$ | 53.41 (↑ 1.46%) | 70.57 (↑ 1.46%) |
| Faster-RCNN | VM | 44.1M | 41.00 | $\mathbb{R}^{512}$ | 40.50 (↑ 5.47%) | 60.30 (↑ 2.20%) |
| Deformable DETR | VM | 41.2M | 41.10 | $\mathbb{R}^{512}$ | 44.10 (↑ 0.68%) | 63.30 (↑ 1.12%) |
| YOLOv7 | VLM | 37.4M | 3.70 | $\mathbb{R}^{512}$ | 51.75 (↑ 3.42%) | 70.12 (↑ 2.15%) |
| YOLOv9 | VLM | 25.5M | 4.16 | $\mathbb{R}^{512}$ | 53.36 (↑ 1.37%) | 70.55 (↑ 1.43%) |
| Faster-RCNN | VLM | 44.1M | 41.02 | $\mathbb{R}^{512}$ | 40.50 (↑ 5.47%) | 60.40 (↑ 2.37%) |
| Deformable DETR | VLM | 41.2M | 41.28 | $\mathbb{R}^{512}$ | 44.40 (↑ 1.37%) | 63.30 (↑ 1.12%) |
| YOLOv7 | LLM | 38.2M | 3.79 | $\mathbb{R}^{1024}$ | 51.36 (↑ 2.64%) | 69.40 (↑ 1.17%) |
| YOLOv9 | LLM | 25.8M | 4.20 | $\mathbb{R}^{1024}$ | 53.28 (↑ 1.22%) | 70.48 (↑ 1.33%) |
| Faster-RCNN | LLM | 44.6M | 41.03 | $\mathbb{R}^{1024}$ | 40.70 (↑ 5.99%) | 60.80 (↑ 3.05%) |
| Deformable DETR | LLM | 41.7M | 41.35 | $\mathbb{R}^{1024}$ | 44.16 (↑ 0.91%) | 63.10 (↑ 1.12%) |

where $z_{i,w,h}^{\text{rd}} \in \mathbb{R}^f$ represents the $i$-th batch output feature of $d$ at position $(w, h)$, $B$ is the batch size, $\tau$ is the temperature parameter, and $\text{sim}(\cdot, \cdot)$ denotes the cosine similarity between two vectors.

This distillation process ensures that, within the specific domain, the atoms in $d$ can effectively approximate various potential linear combinations found in the original **Dictionary** $D$. By selectively removing atoms from RD that are not pertinent to the dataset domain, we achieve a significant reduction in atom count—by at least 50%. This reduction not only increases the model's efficiency but also maintains the performance and expressiveness within the targeted domain.

# 4 EXPERIMENT

## 4.1 SETUPS

**Experimental setup.** We primarily validated the method on the Microsoft COCO dataset (Lin et al., 2014), training on the COCO 2017 train set and evaluating on the COCO 2017 validation set. For Object Detection and Segmentation, we respectively used mAP and mAP@.5 as evaluation metrics, testing on YOLOv7, YOLOv9, Faster RCNN, and Deformable DETR. For the Classification task, we used the CIFAR-100 dataset with the YOLOv9-classify model, using top-1 and top-5 accuracy as metrics.

**Implementation details.** All experiments were conducted using 8 Nvidia V100 GPUs. In the main series of experiments, we trained a modified YOLOv7 model, which included the addition of a **Retriever-Dictionary** Module, for 300 epochs in 2 days. The YOLOv9-based model was trained for 5 days with 500 epochs. We also trained a modified Faster RCNN, based on the mm-detection framework, for a maximum of 120 epochs over 3 days. For Deformable DETR, we trained for approximately 120 epochs over 7 days. The classification task on CIFAR-100 (Krizhevsky et al., 2009) took 2 hours on a single Nvidia 4090 GPU for 100 epochs. Latency was measured on a single Nvidia 3090 GPU without any external acceleration tools, using milliseconds per batch with a batch size of 32 on the MSCOCO validation set.

## 4.2 COMPARISION WITH RD

**Apply to state-of-the-art detectors.** As demonstrated in Table 1, we evaluate the proposed module mainly on YOLOv7 and also integrate it with both fundamental and SOTA real-time object detection models. The ***Dictionary*** is initialized separately by three distinct models: Vision Model (VM) with the YOLOv7 backbone, Vision-Language Model (VLM) employing CLIP, and Large Language Model (LLM) based on GPTv2. Among these, CLIP provides the most significant improvement, likely due to its well-balanced performance across both the vision and language domains.

The RD module consistently yields notable improvements. In YOLOv7 and YOLOv9, the introduction of the module increases the parameter count by less than 1%, yet results in substantial performance gains across key metrics. The improvement is comparable to the performance boost achieved by moving to the next model size (e.g., YOLOv7-x, YOLOv9-e), which typically requires a 100% increase in parameters. For more traditional architectures, we incorporate the RD module into Faster R-CNN with a ResNet-50 backbone He et al. (2016). Despite a modest 2% increase in parameters, the model outperforms Faster R-CNN with a ResNet-101 backbone. Furthermore, we extend the module to a transformer-based architecture, specifically Deformable DETR Zhu et al. (2020) with a ResNet-50 backbone. Similar to the previous results, the RD module yields improvements equivalent to upgrading to the ResNet-101 backbone.

These experiments conclusively demonstrate that leveraging dataset information and incorporating knowledge from VM, VLM, and LLM significantly enhances the performance of a wide range of base models, while requiring only minimal additional parameters.

**Apply to other tasks.** The ***Retriever Dictionary*** (RD) module enhances pixel-level features, and its potential benefits extend beyond detection tasks to include other vision tasks, such as segmentation and classification. To validate this, we conducted segmentation experiments on the MSCOCO dataset and classification experiments on the CIFAR-100 dataset, demonstrating the effectiveness of the RD module across both pixel-level and image-level tasks. Table 2 compares the original YOLO multi-task structure with the one incorporating our proposed module. The results clearly show that the ***Retriever Dictionary*** module upgrades performance across classification, detection, and segmentation tasks, demonstrating its effectiveness in enhancing overall multi-tasking performance.

Table 2: Comparing RD at different tasks.

| Task | Metrics(%) | w/o RD | w/ RD | Improve |
|---|---|---|---|---|
| Classification | Top-1 | 74.86 | 75.70 | ↑ 1.12% |
| | Top-5 | 93.72 | 94.28 | ↑ 0.60% |
| Detection | $mAP^{Box}$ | 50.04 | 51.75 | ↑ 3.42% |
| | $mAP^{Box}_{.5}$ | 68.65 | 69.51 | ↑ 2.15% |
| Segmentation | $mAP^{Seg}$ | 40.53 | 41.56 | ↑ 2.54% |
| | $mAP^{Seg}_{.5}$ | 64.00 | 64.64 | ↑ 1.00% |

Table 3: Comparing with different knowledge-based methods.

| Method | mAP | $mAP_{.5}$ | +Params |
|---|---|---|---|
| baseline | 52.64 | 69.56 | - |
| KD | 52.52 | 69.14 | 57.3M |
| YOLO-World | 51.00 | 67.70 | 66.1M |
| RALF | 51.40 | 68.07 | 37.8M |
| **RD (ours)** | **53.36** | **70.55** | **0.2M** |

**Comparison of knowledge integration methods for YOLO.** Table 3 compares RD with various methods for integrating external knowledge into YOLO. Specifically, knowledge distillation (Hinton et al., 2015) uses YOLOv9-e as the teacher model and YOLOv9-c as the student model, denoted as KD. Another approach, YOLO-World (Cheng et al., 2024), incorporates visual-language concepts from CLIP into YOLO to enhance its understanding of both domains. Additionally, RALF (Kim et al., 2024) utilizes CLIP's text encoder to create a vocabulary set as a database in a RAG-based method. The "+Params" column represents the additional parameters introduced by the knowledge provider or supervisor model compared to the baseline. Overall, RD not only offers the lightest solution, with only 0.2 additional parameters but also delivers the best performance, making it a highly efficient and effective approach.

## 4.3 ABLATION STUDIES

**Fuse coefficient generator and global information exchanger.** In Section 3.1, we discussed splitting the ***Retriever*** core into pointwise convolution **G** and depthwise convolution **E** can significantly reduce the number of parameters. Without a split ***Retriever*** core, the 1-stage coefficient

Table 4: Analysis of fuse *Retriever* core.

| Retriever | Weight size | Params. | mAP |
|---|---|---|---|
| Fuse$(\mathbf{E}, \mathbf{G})(X)$ | $\mathbb{R}^{f \times N \times k \times k}$ | 41.9M | 50.95% |
| $\mathbf{E}(\mathbf{G}(X))$ | $\mathbb{R}^{f \times N}, \mathbb{R}^{N \times k \times k}$ | 4.3M | 50.64% |

Table 5: Performance comparison of different models on the COCO 2017 validation set.

| BackBone | Initializer | algorithm | Params | Atoms | Feature | mAP$^{val}_{.5}$ | mAP$^{val}_{.5:.95}$ |
|---|---|---|---|---|---|---|---|
| YOLOv7 | Random | - | 37.4M | 512 | $\mathbb{R}^{512}$ | 50.02% | 69.06% |
| YOLOv7 | YOLOv7 | - | 37.4M | 512 | $\mathbb{R}^{512}$ | 51.34% | 69.37% |
| YOLOv7 | CLIP | - | 37.4M | 512 | $\mathbb{R}^{512}$ | 51.75% | 70.12% |
| YOLOv7 | GPT | - | 38.2M | 512 | $\mathbb{R}^{1024}$ | 51.33% | 69.40% |
| YOLOv7 | YOLOv7 | $k$-means | 37.4M | 512 | $\mathbb{R}^{512}$ | 51.34% | 69.37% |
| YOLOv7 | YOLOv7 | VAE | 37.4M | 512 | $\mathbb{R}^{512}$ | 51.19% | 69.25% |
| YOLOv7 | CLIP | $k$-means | 37.4M | 512 | $\mathbb{R}^{512}$ | 51.75% | 70.12% |
| YOLOv7 | CLIP | Convex Hull | 37.8M | 1024 | $\mathbb{R}^{512}$ | 49.58% | 68.92% |
| YOLOv7 | GPT | Normalize | 38.2M | 512 | $\mathbb{R}^{1024}$ | 51.33% | 69.40% |
| YOLOv7 | GPT | Tanh | 38.2M | 512 | $\mathbb{R}^{1024}$ | 51.27% | 69.35% |
| YOLOv7 | CLIP | - | 37.4M | 512 | $\mathbb{R}^{512}$ | 51.75% | 70.12% |
| YOLOv7 | GPT | - | 38.2M | 512 | $\mathbb{R}^{1024}$ | 51.33% | 69.40% |
| YOLOv7 | Mix | {LLM, VLM} | 37.4M | 512 | $\mathbb{R}^{512}$ | 51.38% | 69.59% |

generates the process as follows:

$$\mathrm{R}'(X) = \text{Fuse}\,(\mathbf{E}, \mathbf{G})(X) = X_{w,h} * W^{eq} \tag{9}$$

where $W^{eq} \in \mathbb{R}^{k \times k \times f \times n}$ is the convolution matrix, $\mathrm{R}' : \mathbb{R}^{f \times W \times H} \rightarrow \mathbb{R}^{N \times W \times H}$ is the 1-stage generate process. This operation will require an extra of $Nfk^2 - (fN + Nk^2)$ parameters. And the computational complexity is of $O(WHNfk^2)$, compared to $\mathbf{G}$ and $\mathbf{E}$, whose complexity is of $O\left(WH(fN + Nk^2)\right)$, $\mathrm{R}'$ requires a huge amount of operations and parameters, but only receives slightly better performance, as demonstrated in Table 4.

**Different *Dictionary* construction strategies.**  Table 5 presents an ablation study on more different strategies for constructing the *Dictionary*. For VM, a VAE (Kingma & Welling, 2014) is used to capture the dataset's general distribution and build the *Dictionary*. In VLM, we leverage whole CLIP (Radford et al., 2021)'s image encoder for global embeddings, as outlined in Section 3.2.

Additionally, we explore using a convex hull of the dataset's feature distribution to address outliers, though this limits the RD-module's ability to represent common features. For LLM, normalization techniques like tanh and standard normalization are applied to manage outliers. While tanh compresses large feature distances, standard normalization is more effective in handling outliers.

Finally, we combine feature distributions from LLM and VLM to create a blended representation. Despite yielding positive results, these methods fail to produce a uniform distribution and underperform compared to the strategy described in the methodology.

## 4.4 VISUALIZATION

In this section, we employ visualization to demonstrate the impact of the RD-module on object detection models and to visualize the *Retriever*'s selection of atoms from the *Dictionary*.

**Visualization of *Dictionary* atom coefficients.**  To gain a deeper understanding of the behavior of the *Retriever* and *Dictionary* core, we visualized the atom coefficients and their distribution during the forward pass. In Figure 4b, the X-axis represents the correlation with the current input and each

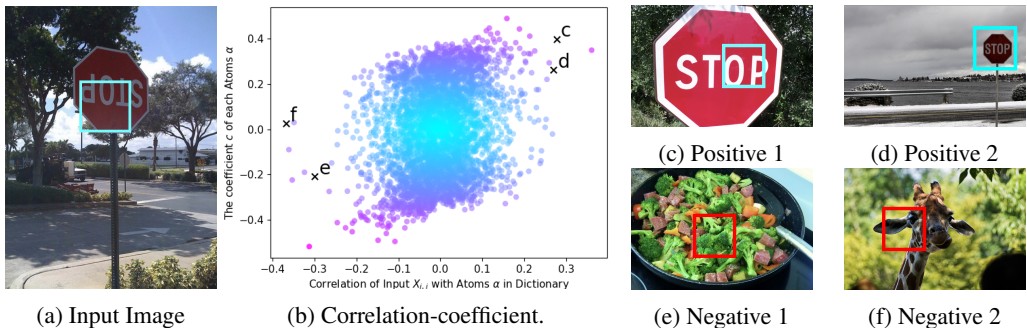

(a) Input Image      (b) Correlation-coefficient.      (c) Positive 1      (d) Positive 2

(e) Negative 1      (f) Negative 2

Figure 4: Visualization of Dictionary Atom Coefficients

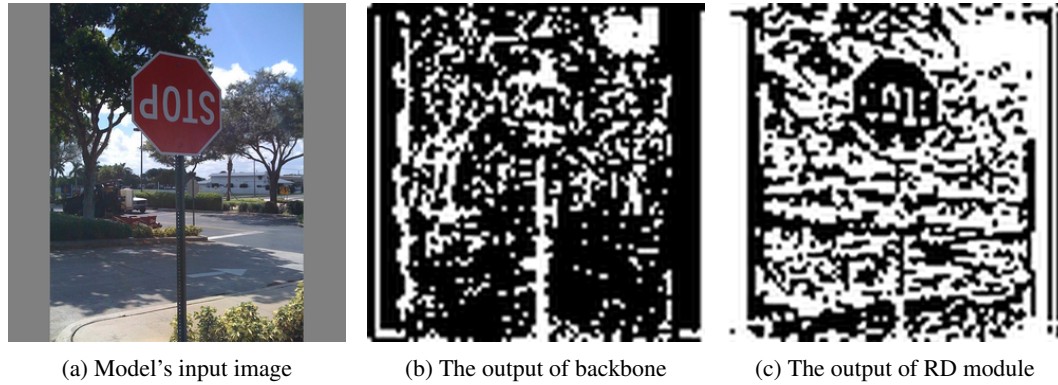

(a) Model's input image      (b) The output of backbone      (c) The output of RD module

Figure 5: Visualization of w/o RD module

$\alpha$, while the Y-axis displays the corresponding coefficient for each atom. The surrounding points retrieved from the dataset, relevant to the current input, are marked as $\times$ on the plot.

Using the bounding box area in Figure 4a as the input, we generate a correlation-coefficient map. As depicted in Figure 4b, most high-correlation and high-coefficient pairs (Figures 4c,4d) correspond to traffic signs, while the low-correlation and low-coefficient pairs (Figures 4e,4f) do not. This visualization demonstrates that the *Retriever Dictionary* effectively selects the relevant atoms to enhance input features while attenuating non-relevant atoms, thereby reinforcing the input's key characteristics.

**Visualizing model backbone output with and without *Dictionary*.** Using the same traffic signals as the model input (Figure 5a), we visualize the feature map generated by the origin backbone network (Figure 5b) and the backbone output including the RD-module (Figure 5c). In Figure 5c, the Traffic Sign pattern is distinctly retained, and background information is preserved. In contrast, Figure 5b retains only parts of the railings and the input photo padding block. This comparison demonstrates that the RD-module helps the model retain important information while eliminating unimportant details.

## 5 CONCLUSION

The *Retriever Dictionary* module offers a lightweight and efficient approach to incorporating dataset knowledge into YOLO through various modality models. By leveraging pre-stored explicit knowledge within the *Dictionary*, the *Retriever* effectively retrieves relevant information while the Dictionary learning mechanism enables fine-tuning of the atoms. This module demonstrates its versatility, providing improvements not only in YOLO-based tasks but also across a range of foundational object detection models and broader computer vision tasks. We believe that this work lays the foundation for further exploration of real-time computer vision models that integrate explicit or external knowledge sources.

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

## A APPENDIX

### A.1 REPRODUCIBILITY AND TRAINING SETUP

In Table 5, we observed that some of the mAP values for the original YOLOv7 and YOLOv9 models differ from those reported in the original papers. As noted in the corresponding GitHub issue, this discrepancy may be attributed to differences in the number of GPUs used, as well as the reduced effectiveness of batch normalization when training with smaller batch sizes. To ensure a more consistent and fair comparison, we followed the official code instructions and re-trained the models on our hardware setup, using the same batch sizes and number of GPUs specified in the original papers. Despite adhering to the official training guidelines, our re-run results for YOLOv7 and YOLOv9 produced slightly lower mAP values than those reported in the original papers. Nevertheless, our implementation of the *Retriever Dictionary* (RD) method is still higher than those reported in the original papers, demonstrating the effectiveness of our approach.

For the Deformable DETR and Faster R-CNN models, we used the MMDetection training code. In this case, our results were consistent with the values reported in the original papers, likely due to the fact that MMDetection provides a standardized training batch configuration, minimizing the impact of hardware differences on model performance.

### A.2 ARCHITECTURES OF THE MODEL WITH *Retriever Dictionary*

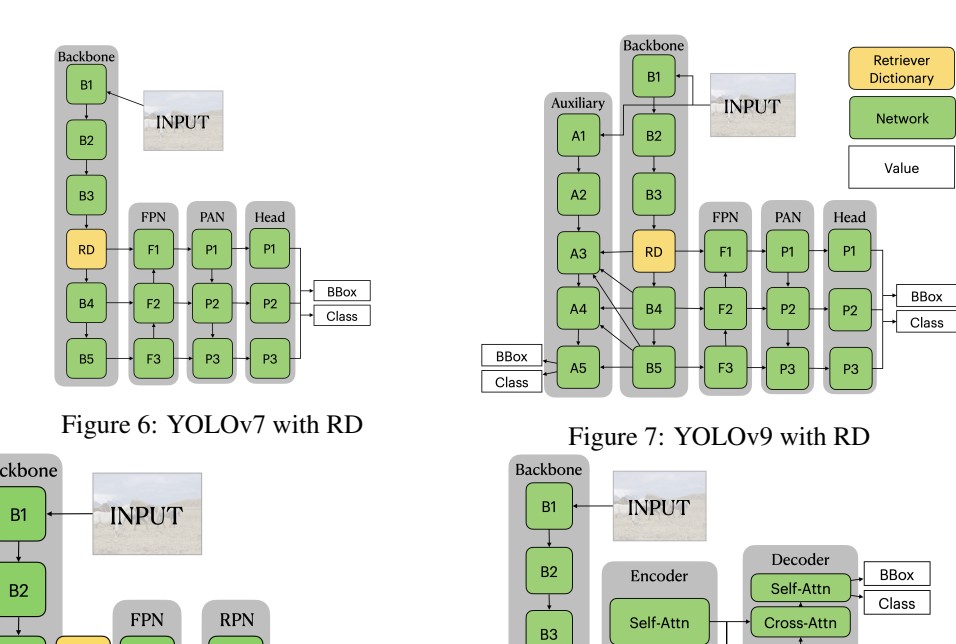

Figure 6: YOLOv7 with RD

Figure 7: YOLOv9 with RD

Figure 8: Faster-RCNN with RD

Figure 9: Deformable DETR with RD

### A.3 ADDITIONAL EXPERIMENT RESULTS

**Detailed mAP Results for All Experiments.** Table 6 provides the detailed mAP values for each experiment discussed in the experiment section. It includes metrics such as AP, $AP_{.5}$, $AP_{.75}$, $AP_S$, small object; $AP_M$, middle object; and $AP_L$, large object.

Table 6: Comprehensive mAP Metrics Across Experiments

| BackBone | Initializer | Params | AP | AP$_{.5}$ | AP$_{.75}$ | AP$_S$ | AP$_M$ | AP$_L$ |
|---|---|---|---|---|---|---|---|---|
| YOLOv7 | VM | 37.4M | 51.34 | 69.37 | 56.42 | 35.86 | 56.10 | 66.78 |
| YOLOv7 | VLM | 37.4M | 51.75 | 70.12 | 56.70 | 36.00 | 57.12 | 66.32 |
| YOLOv7 | LLM | 38.2M | 51.33 | 69.40 | 56.40 | 34.72 | 56.70 | 67.13 |
| YOLOv7 | VAE | 37.4M | 51.34 | 69.37 | 56.42 | 35.86 | 56.10 | 66.78 |
| YOLOv7 | Convex | 37.4M | 49.58 | 68.92 | 55.60 | 34.72 | 55.14 | 66.02 |
| YOLOv7 | Tanh | 37.4M | 51.27 | 69.35 | 56.28 | 35.51 | 56.34 | 66.09 |
| YOLOv7 | Mix | 37.4M | 51.37 | 69.49 | 56.26 | 35.17 | 56.43 | 66.94 |
| YOLOv9 | VM | 25.8M | 53.41 | 70.57 | 58.40 | 36.29 | 58.93 | 69.94 |
| YOLOv9 | VLM | 25.8M | 53.36 | 70.55 | 57.98 | 36.39 | 59.04 | 69.95 |
| YOLOv9 | LLM | 25.8M | 53.28 | 70.48 | 57.81 | 35.91 | 58.80 | 69.55 |
| Faster-RCNN | VM | 44.1M | 40.50 | 60.30 | 44.20 | 24.90 | 43.20 | 51.80 |
| Faster-RCNN | VLM | 44.1M | 40.50 | 60.40 | 44.40 | 25.10 | 43.30 | 52.50 |
| Faster-RCNN | LLM | 44.6M | 40.70 | 60.80 | 44.70 | 25.60 | 43.70 | 52.60 |
| Deformable DETR | VM | 41.2M | 44.10 | 63.00 | 48.20 | 26.30 | 47.00 | 59.00 |
| Deformable DETR | VLM | 41.2M | 44.40 | 63.30 | 48.30 | 26.50 | 47.90 | 58.70 |
| Deformable DETR | LLM | 41.7M | 44.20 | 63.10 | 47.80 | 26.20 | 47.60 | 58.70 |
| YOLOv7 | baseline | 37.2M | 50.04 | 68.95 | 55.10 | 34.20 | 55.70 | 66.20 |
| YOLOv7-x | baseline | 71.3M | 51.29 | 70.27 | 56.78 | 35.84 | 56.63 | 67.59 |

**Comparison of Frozen vs. Fully Trained *Dictionary* Strategies.** Table 7 compares two training strategies: one where only the model $\mathcal{B}$ and *Retriever* are trained while the *Dictionary* remains frozen, and another where the model, *Retriever*, and *Dictionary* are all fully trained. In the table, ✓ indicates that the corresponding component is trainable. Table 7a shows results using pre-trained weights, with row 3 displaying the fine-tuning of the original model. Table 7b reports results from training the model from scratch. The full training strategy slightly outperforms the frozen *Dictionary* method in both scenarios, with both approaches surpassing the performance of the original model. These findings highlight fine-tuning the *Dictionary* effectively helps the model's output distribution.

Table 7: Comparison of Model Performance with and without Freezing $D$ during Training.

(a) Fine-tune with pre-trained weight.

| $\mathcal{B}$ | $R$ | $D$ | AP$_{.5:.95}$ | AP$_{.5}$ |
|---|---|---|---|---|
| ✓ | ✓ | ✓ | 52.73 | 69.61 |
| ✓ | ✓ | | 52.64 | 69.57 |
| ✓ | - | - | 51.66 | 68.12 |

(b) Training from scratch.

| $\mathcal{B}$ | $R$ | $D$ | AP$_{.5:.95}$ | AP$_{.5}$ |
|---|---|---|---|---|
| ✓ | ✓ | ✓ | 51.72 | 70.12 |
| ✓ | ✓ | | 51.35 | 69.52 |

**Performance Improvements in Classification with RD Module.** Table 8 demonstrates the performance improvement of the *Retriever Dictionary* (RD) module in the classification task. In the YOLO classification task, YOLOv8 employs CSPNet (Wang et al., 2020) as the backbone, while YOLOv9 uses GELAN as the backbone. We tested both backbones with our RD module, and the results show that the RD module provides performance improvements in both structures.

Table 8: More classification task.

| RD | Model | Epoch | Top-1 | Top-5 |
|---|---|---|---|---|
| | ELAN | 100 | 71.85 | 92.93 |
| ✓ | ELAN | 100 | **74.18** | **93.14** |
| | GELAN | 100 | 74.86 | 93.72 |
| ✓ | GELAN | 100 | **75.70** | **94.28** |

Table 9: Transfer learning on small dataet.

| RD | Pretrained | Epoch | mAP(%) | mAP$_{.5}$(%) |
|---|---|---|---|---|
| | ✓ | 10 | 88.48 | 65.87 |
| ✓ | ✓ | 10 | **91.54** (↑ 3.46%) | **74.63** (↑ 13.30%) |
| | ✓ | 100 | 92.28 | 76.79 |
| ✓ | ✓ | 100 | **92.93** (↑ 0.70%) | **78.02** (↑ 1.60%) |
| | | 100 | 84.44 | 65.49 |
| ✓ | | 100 | **85.15** (↑ 0.84%) | **66.33** (↑ 1.28%) |

## A.4 Transfer Learning with *Retriever Dictionary* on VOC Dataset

In Table 9, we demonstrate the effectiveness of the ***Retriever Dictionary*** (RD) on a transfer learning task. Using pre-trained weights from the MSCOCO dataset, we trained the model on the VOC (Everingham et al., 2010) dataset with three learning rate schedules: 10-epoch fast training, 100-epoch full-tuning, and training from scratch. In the fast training scenario, the model with the RD module showed significantly faster convergence compared to the model without the module. In the full-tuning scenario, the RD-enhanced model achieved higher performance. Lastly, in the training from scratch scenario, our RD module provided the model with better information, yielding superior results even on a smaller dataset.

## A.5 Further Visualizations: Original vs. RD-Module Models

We present additional examples in Figures 10 and 11, illustrating input images, the outputs from the original model, and the outputs from the model with the RD module. The results demonstrate that the RD-Model outputs are noticeably clearer. For example, in Figure 10 (ID 1 and 2), the edges of objects are significantly sharpened. Similarly, in Figures 10 and 11 (ID 3, 4, 5, and 6), our model exhibits higher accuracy and fewer false positives in the object's bounding boxes, as indicated by the red arrows.

## A.6 Pseudo code of full training process of *Retriever-Dictionary* Model

The complete training process, from initialization to final model, follows the pseudo-code provided in Algorithm 1. This process includes ***Dictionary*** initialization, regular model training, and ***Dictionary*** compression. The overall training time is approximately equivalent to the original training epochs, with an additional 2 epochs allocated for setup and compression.

## A.7 Visualization of Initial Distributions Across Different Modality Models

Figure 12 and 13 visualizes the t-SNE distributions of VM, VLM, and LLM dictionaries. Vision and Language dictionaries occupy distinct regions, while Vision-Language overlaps with Vision. Notably, the Vision-Language dictionary is more uniformly distributed, showcasing its ability to provide richer information.

## A.8 Deeper Discussion of the *Retriever* Core

**Two Convolutions Without Activation Functions.** Consider two consecutive $1 \times 1$ convolutional layers without activation functions. The first layer has weights $W^1 \in \mathbb{R}^{M \times N}$, and the second has weights $W^2 \in \mathbb{R}^{N \times M}$, with an input $\mathbf{X} \in \mathbb{R}^{N \times H \times W}$.

For the first convolutional layer, the output at a spatial location $(h, w)$ is defined as:

$$\mathbf{Z}^1_{m,h,w} = \sum_{n=1}^{N} W^1_{m,n} \cdot \mathbf{X}_{n,h,w},$$

where $W^1$ is the weight of the first convolution, $n \in [0, N)$ is the channels, and the second convolution's output is as follows:

$$\begin{aligned}
\mathbf{Z}^2_{n,h,w} &= \sum_{m=1}^{M} W^2_{n,m} \cdot \mathbf{Z}^1_{m,h,w} \\
&= \sum_{m=1}^{M} W^2_{n,m} \cdot \left( \sum_{n=1}^{N} W^1_{m,n} \cdot \mathbf{X}_{n,h,w} \right) \\
&= \sum_{n=1}^{N} \left( \sum_{m=1}^{M} W^2_{n,m} \cdot W^1_{m,n} \right) \cdot \mathbf{X}_{n,h,w}.
\end{aligned}$$

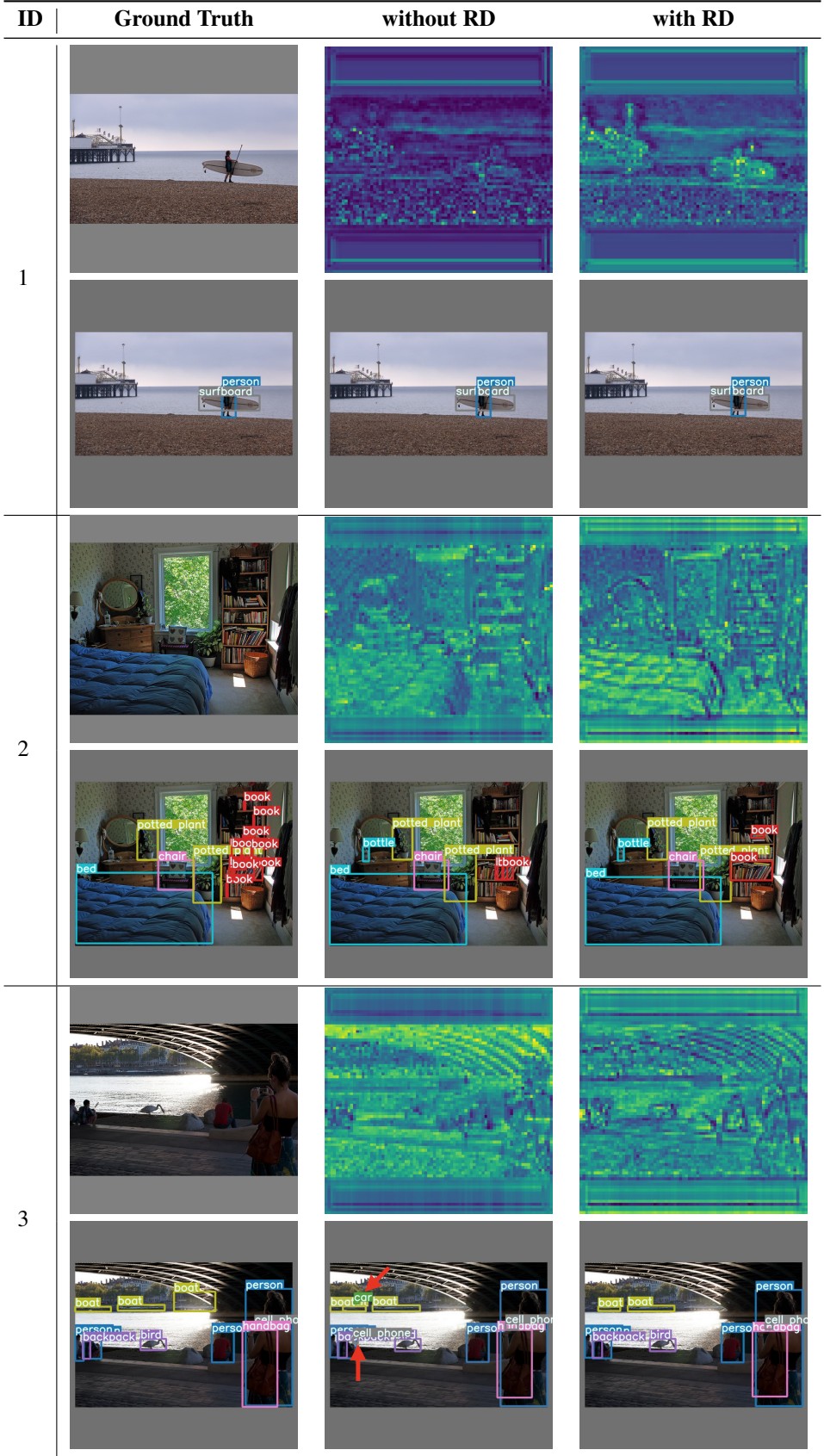

Figure 10: More visualization of RD

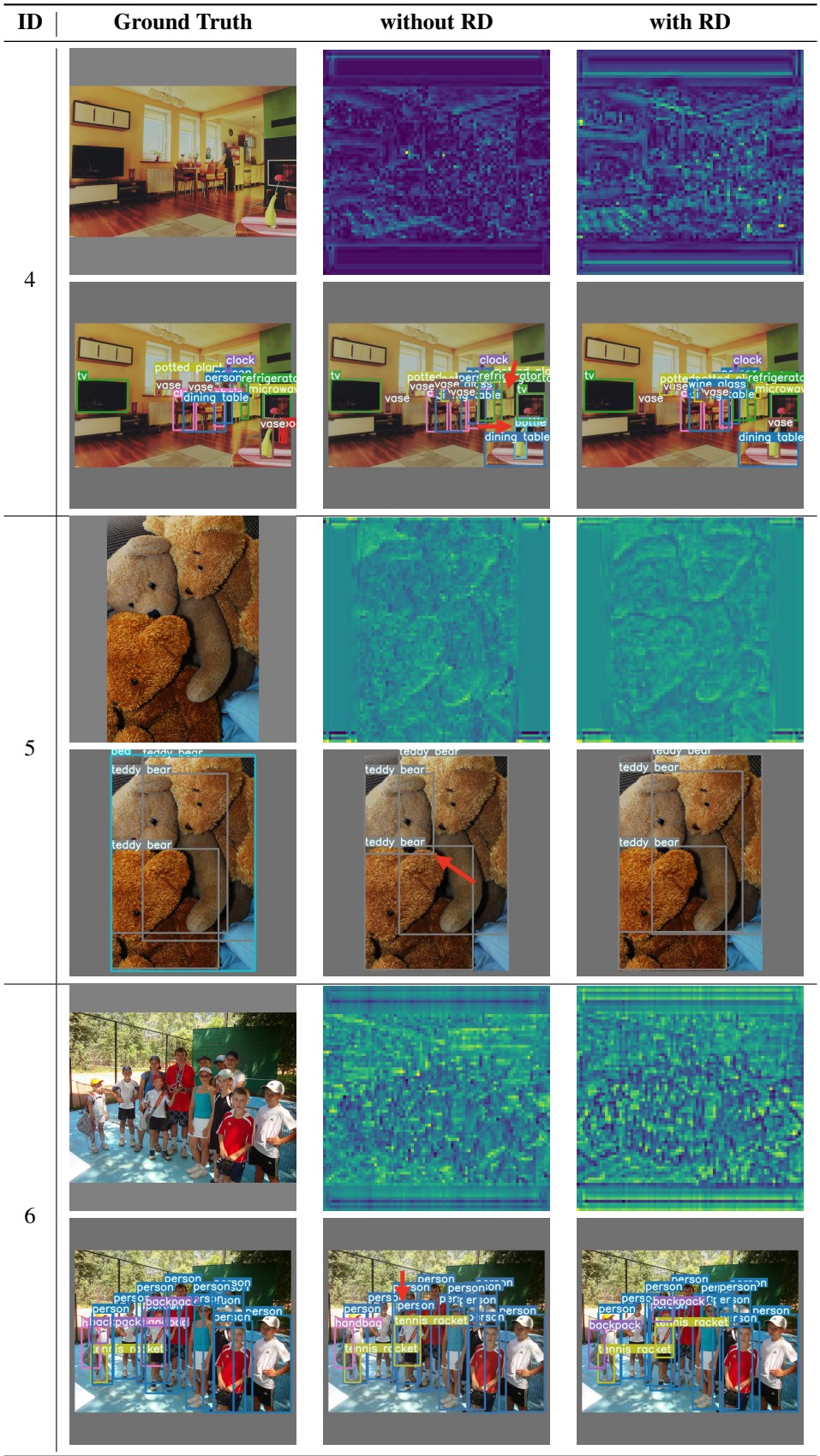

Figure 11: More visualization of RD

---

**Algorithm 1:** Train a model with ***Retriever Dictionary***

---

**Data:** Dataset with images and bounding boxes
**Result:** Trained model with ***Retriever Dictionary***
   // Initialization of the ***D***ictionary
1 **foreach** $(img, box) \in \mathrm{D}ataset$ **do**
2      $features \leftarrow$ encoder$(img)$
3 RD $\leftarrow$ new ***Retriever-Dictionary***(kmeans($features$))
4
   // Standard Training Method
5 backbone, head $\leftarrow$ new Model()
6 **for** *epoch* $e = 1$ **to** *num_epochs* **do**
7      **foreach** $(img, box) \in \mathrm{D}ataset$ **do**
8           $output \leftarrow$ head(RD(backbone($img$)))
9           $loss \leftarrow$ loss_function($output$, $box$)
10           update($loss$, (backbone, RD, head))
11
   // ***D***ictionary Compression
12 rd $\leftarrow$ new ***Retriever-Dictionary***(choice_from($D$))
13 **foreach** $(img, box) \in \mathrm{D}ataset$ **do**
14      freeze(backbone)
15      $teacher\_feature \leftarrow$ RD(backbone($img$)))
16      $student\_feature \leftarrow$ rd(backbone($img$)))
17      $loss \leftarrow$ cosine_similarity($teacher\_feature$, $student\_feature$)
18      update($loss$, rd)
19
   // Final Model:
20 FullModel $\leftarrow$ merge(backbone, rd, head)

---

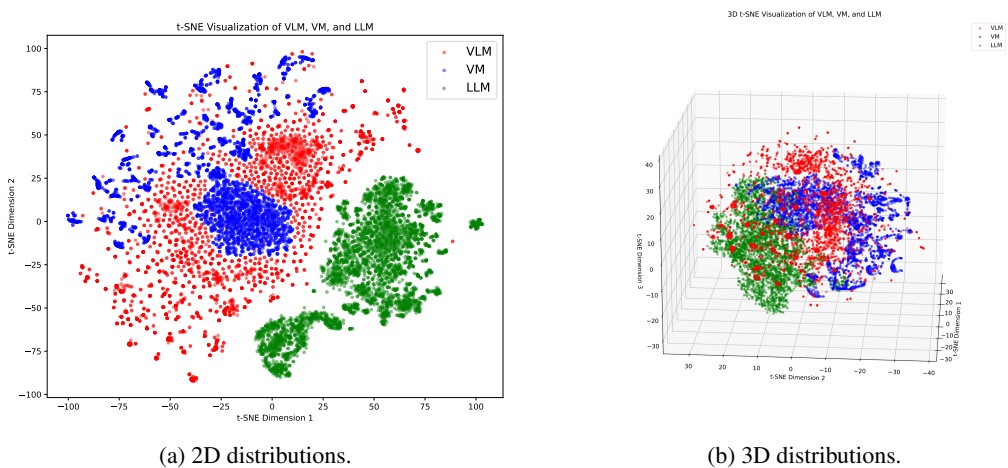

(a) 2D distributions.            (b) 3D distributions.

Figure 12: The initial distributions of dictionaries derived from different modality models.

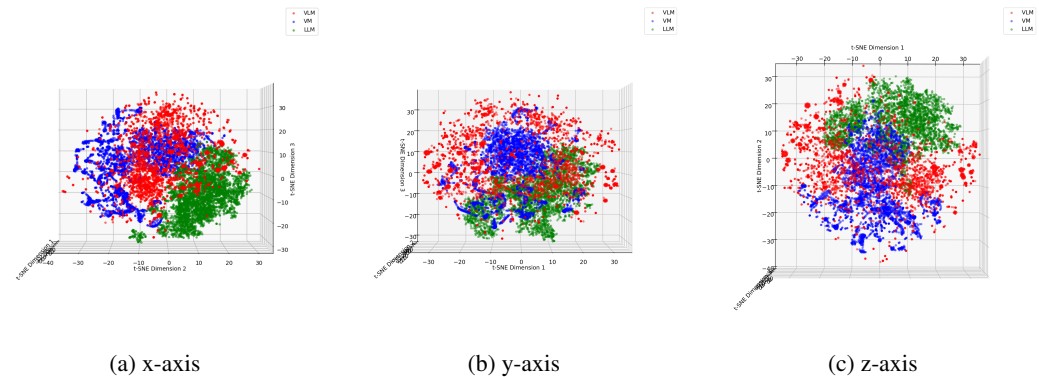

(a) x-axis               (b) y-axis              (c) z-axis

Figure 13: Orthographic projections of the initial distributions using 3D t-SNE. Each subfigure represents a different axis projection to better illustrate the structure of the distributions.

Thus, the equivalent weight matrix is given by:

$$W_{n,n'}^{\text{eq}} = \sum_{m=1}^{M} W_{n,m}^2 \cdot W_{m,n'}^1, \quad \forall n, n' \in N.$$

The original two-layer architecture has $2NM$ parameters. The equivalent layer has $N^2$ parameters. Without an activation function, the number of parameters in the combined layer is wasted by $2NM - N^2$. Even if $M < \frac{N}{2}$, the rank of $W^1 W^2$ will be limited by $\min(M, N)$, leading to a significant drop in abilities of the network. Therefore, activation functions are essential in most cases to maintain the representational capacity of the sequential convolutional layers.

**Forward Pass with the *Retriever* Core.** We now extend this discussion to our *Retriever* core, consisting of two convolutional layers: a pointwise ($1 \times 1$) convolution for channel projection and a depthwise convolution, with the number of groups equal to the number of channels. And there is no activation function between layers.

For the pointwise convolution (first layer), we compute:

$$\mathbf{Y}_{c,h,w} = \sum_{i=1}^{f} W_{c,i}^G \cdot \mathbf{X}_{i,h,w}, \tag{10}$$

where the $W_{c,i}^G$ is the pointwise convolution weight, as well as the Coefficient Generator weight. For the depthwise convolution (second layer), the operation is given by:

$$\mathbf{Z}_{c,h,w} = \sum_{m=-k}^{k} \sum_{n=-k}^{k} W_{c,m,n}^E \cdot \mathbf{Y}_{c,h+m,w+n}, \tag{11}$$

where the $W_{c,m,n}^E$ is the depthwise convolution weight, as well as Global Information Exchanger, $c$ is the channel dimension. Since no non-linear operation is applied between these two layers, we can combine them into a single equivalent convolution. Substituting the output of the first layer into the second, we get:

$$\mathbf{Z}_{c,h,w} = \sum_{m=-k}^{k} \sum_{n=-k}^{k} W_{c,m,n}^{E} \cdot \left( \sum_{i=1}^{f} W_{c,i}^{G} \cdot \mathbf{X}_{i,h+m,w+n} \right) \tag{12}$$

$$= \sum_{i=1}^{f} W_{c,i,0,0}^{G} \cdot \left( \sum_{m=-k}^{k} \sum_{n=-k}^{k} W_{c,m,n}^{E} \cdot \mathbf{X}_{i,h+m,w+n} \right) \tag{13}$$

$$= \sum_{i=1}^{f} \sum_{m=-k}^{k} \sum_{n=-k}^{k} W_{c,i,0,0}^{G} \cdot W_{c,m,n}^{E} \cdot \mathbf{X}_{i,h+m,w+n}. \tag{14}$$

This results in a combined convolution operation:

$$W_{c,i,m,n}^{\mathrm{eq}} = W_{c,i,0,0}^{G} \cdot W_{c,i,m,n}^{E},$$

where $c$, $i$, $m$, and $n$ represent the input channels, output channels, and two spatial dimensions of the kernels, respectively. $W_{c,i,0,0}^{G}$ denotes the pointwise convolution (with both spatial indices set to 0). The resulting equivalent kernel has a size of $k \times k$, with equivalent weights $W^{\mathrm{eq}} \in \mathbb{R}^{N \times f \times k \times k}$, where $f$ is the input feature dimension and $N$ is the number of output channels. While this equivalent convolution maintains the functionality of the original two layers, the combined weights are computationally heavier due to the different dimensions involved (channel dimension and kernel size). Nevertheless, even without an activation function, the model behaves as a normal convolution operation.

**Gradient Descent and Weight Update.** Following the forward pass, the weight update rules for the pointwise and depthwise convolutional layers can be expressed as:

$$W^{G'} = W^{G} - \eta \frac{\partial L}{\partial W^{G}}, \quad W^{E'} = W^{E} - \eta \frac{\partial L}{\partial W^{E}},$$

where $eta$ represents the learning rate, and $W^{G'}$ and $W^{G'}$ denote the updated weights for the pointwise and depthwise convolutions, respectively. *Retriever* core update is then given by:

$$W^{G'} \cdot W^{E'} = \left( W^{G} - \eta \frac{\partial L}{\partial W^{G}} \right) \cdot \left( W^{E} - \eta \frac{\partial L}{\partial W^{E}} \right)$$

$$= W^{G} \cdot W^{E} - \eta \left( W^{G} \cdot \frac{\partial L}{\partial W^{E}} + \frac{\partial L}{\partial W^{G}} \cdot W^{E} \right) + \eta^{2} \frac{\partial L}{\partial W^{G}} \cdot \frac{\partial L}{\partial W^{E}}.$$

Since $\eta$ (the learning rate) is generally small during training, especially when using FP16 precision, the second-order term can be ignored. The simplified weight update for the equivalent kernel becomes:

$$W^{G'} \cdot W^{E'} \approx W^{G} \cdot W^{E} - \eta \left( W^{G} \cdot \frac{\partial L}{\partial W^{E}} + \frac{\partial L}{\partial W^{G}} \cdot W^{E} \right) = W_{\mathrm{eq}} - \eta W^{G} \cdot \frac{\partial L}{\partial W^{E}} + \frac{\partial L}{\partial W^{G}} \cdot \eta W^{E}.$$

This follows the structure of the Taylor expansion:

$$f(x + \delta x, y + \delta y) \approx f(x, y) + \frac{\partial f}{\partial x} \delta x + \frac{\partial f}{\partial y} \delta y, \tag{15}$$

this gives the updated equivalent weight, which closely approximates the equivalent convolution:

$$W_{\mathrm{eq}}' = W_{\mathrm{eq}} - \eta \frac{\partial L}{\partial W_{\mathrm{eq}}} \approx W^{G'} W^{E'}.$$

A.9  NOTATIONS

Table 10: Notation Reference Table for Symbols Used in the Paper

| Notation | Default Value | Description |
|---|---|---|
| $D$ | - | The *Dictionary*, composed of learned atoms |
| $N$ | 512 | Number of atoms $\alpha$ in the *Dictionary* $D$ |
| $\alpha$ | - | Each individual element (atom) in the *Dictionary* $D$ |
| $\alpha_i$ | - | The $i$-th atom in the *Dictionary* $D$ |
| $W$ | 80 | The width dimension of the input to the RD module |
| $H$ | 80 | The height dimension of the input to the RD module |
| $X$ | - | Input feature map to the RD module |
| $X_{h,w}$ | - | The pixel value at position $(h,w)$ of the input $X$ |
| $f$ | 512 | Dimensionality of each atom in the *Dictionary* |
| $k$ | 5 | Global Information Exchanger kernel size |
| $c$ | - | Coefficient matrix before normalized used to weight atoms in the RD module |
| $c_{i,h,w}$ | - | Coefficient value before normalized for the $i$-th atom in the *Dictionary* $D$, at pixel $(h,w)$ |
| $c'_{i,h,w}$ | - | Normalized coefficient value for the $i$-th atom in the *Dictionary* $D$, at pixel $(h,w)$ |
| $z^{\phi}_{i,h,w}$ | - | The feature of generate by backbone and module $\phi$ for the $i$-th mini batch, at pixel $(h,w)$ |
| $\lambda$ | 0.8 | Residual weight in the forward pass of the RD module |
| $\mathbf{G}(\cdot)$ | - | Coefficient Generator function in the RD module |
| $W^G$ | - | Convolutional matrix used in the Coefficient Generator $\mathbf{G}(\cdot)$ |
| $\mathbf{E}(\cdot)$ | - | Global Information Exchanger function in the RD module |
| $W^E$ | - | Convolutional matrix used in the Global Information Exchanger $\mathbf{E}(\cdot)$ |

