# OpenReview forum: "YOLO-RD: Introducing Relevant and Compact Explicit Knowledge to YOLO by Retriever-Dictionary"
_ICLR.cc/2025/Conference — ICLR 2025 Poster_

### Official Review · Reviewer_hT7W · 2024-10-30

**Soundness:** 3
**Presentation:** 2
**Contribution:** 3
**Rating:** 6
**Confidence:** 4

**Summary:**

The paper purposed a Retriever-Dictionary (RD) module to solve the balancing problem between current input and whole dataset information in object identification and localization. Its retriever processes region features to generate a query, and the dictionary that saves dataset information could prompt the query to select relevant atoms. The dictionary incorporates different modalities as encoder, ensuring a good overall understanding of the dataset in various modality spaces.

**Strengths:**

1. **Rigorousity of pipeline design.** The whole Retriever Dictionary design is theoretically robust. The authors have considered the overall operation of Retriever and designed Global Information Exchanger with Coefficient Generator. The introduction of PONO also helps to prevent Dictionary from collapsing into an identity matrix.

2. **Innovative incorporation of multimodal external knowledge in dictionary building.** The authors purposed that the dictionary could be initialized by multiple modalities of models, ensuring a comprehensive representation of the dataset. In response to huge amount of feature vectors involved, operations including k-means and contrasive learning distillation have made the dictionary condensed and efficient.

3. **Congeneric methodology design.** The framework is extendable to RCNNs, Transformer-based detection models and other kindred models. The authors have incorporated experiments on these models. And their experiments supports their theory basically well.

**Weaknesses:**

1. The paper's figure lacks generalization for the whole work, i.e. Figure 1 and 2 cannot fully describe the complete functionality of the RD module. From the dictionary and module structural description, the whole pipeline is hard to follow as the authors have mentioned

2. The authors refer to RAG in their related work and mention its limitations in vision applications. RAG is a system that relies on external multi-modules and is mainly applied to Large language models.The article tries to compare its consistency with RAG in terms of knowledge base ideas, but lacks further explanations on this. e.g. How exactly does it differ from a RAG as part of the network structure?

3. Fig.5, the role of the RD module could have a richer and more well annotated picture explanation. Depending on the modality of initialisation (VM, VLM, LLM), there is also a strong need to visualize the effect of different modalities for initialization.

4. Although the model introduces minimal parameter increase, there is limited discussion on its impact on computational speed and inference latency, especially for deployment in real-time applications.

5. The dictionary initialization process, particularly when using VLMs or LLMs, is not straightforward and may demand significant computational resources during setup. More details on optimizing or reducing this process would be beneficial.

6. The module's efficiency in managing larger datasets or dynamic, continuously updated dictionaries remains unclear. Further analysis on handling diverse data scales could improve its adaptability.

**Questions:**

Have you tried incorporating multple modality initialization and alignment for the dictionary? For example, VM and VLM initialization together? For more questions, please refer to the weaknesses listed above

---

> ### Author Response · Authors · 2024-11-16
>
> > **Q1**: Have you tried incorporating multiple modality initialization and alignment for the dictionary? For example, VM and VLM initialization together? For more questions, please refer to the weaknesses listed above
>
> Yes, we have already explored this. As shown in Table 5 (last row), we incorporated mixing LLM and VLM and we discussed this in Section 4.3 (lines 473-475).
>
> ----
>
> > **W1**: The paper's figure lacks generalization for the whole work, i.e. Figure 1 and 2 cannot fully describe the complete functionality of the RD module. From the dictionary and module structural description, the whole pipeline is hard to follow as the authors have mentioned
>
> Thank you for pointing this out.
> To address this concern, we have revised Figure 2 to offer a more general representation of the RD module, focusing on its broader functionality rather than solely on internal mechanisms. Additionally, we have already included a pseudo-code in Appendix A.6 to provide a clearer understanding of the overall pipeline.
> Figure 1 is designed to illustrate the overarching concept, highlighting how the compact knowledge dictionary effectively integrates external information to address the inefficiencies in leveraging dataset-wide knowledge in prior architecture.
>
> ----
>
> > **W2**: The authors refer to RAG in their related work and mention its limitations in vision applications. RAG is a system that relies on external multi-modules and is mainly applied to Large language models.The article tries to compare its consistency with RAG in terms of knowledge base ideas, but lacks further explanations on this. e.g. How exactly does it differ from a RAG as part of the network structure?
>
> Thank you for pointing out this, we have made a comparison based on your suggestion:
> Differences:
> 1. **Dictionary Learning:** Our RD uses dictionary learning to store and fine-tune information, unlike most RAG, which retrieves fixed data.
> 2. **Feature Representation:** Our RD integrates high-dimensional features for efficient knowledge delivery, while RAG relies on data like sentences, which can be less compact.
> 3. **Efficiency:** Our RD offers low computational overhead and high efficiency, while RAG often requires significant resources for retrieval and processing.
>
> Shared Features:
> 1. Both use external modules for additional knowledge integration.
> 2. Both allow for replacing the dictionary (or database) to update knowledge.
>
> ----
>
> > **W3**: Fig.5, the role of the RD module could have a richer and more well-annotated picture explanation. Depending on the modality of initialisation (VM, VLM, LLM), there is also a strong need to visualize the effect of different modalities for initialization.
>
> - Better Picture Explanation: Yes, we have included additional examples in Appendix A.5, Figures 11 and 12, where we discuss various scenarios with detailed visual explanations.
> - Visualization of Modality Initializations: In Appendix A.7 and Figure 13, we added a new section employing t-SNE to visualize the initialization distributions across different modalities.
>
> As shown in Figure 13, the Language dictionary and Vision dictionary occupy distinct but similarly sized spaces, while the Vision-Language dictionary overlaps with the Vision dictionary. Notably, the Vision-Language dictionary offers a more uniform and well-distributed representation compared to the others, effectively showcasing its ability to provide comprehensive information.
>
> ----
>
> > **W4**: Although the model introduces minimal parameter increase, there is limited discussion on its impact on computational speed and inference latency, especially for deployment in real-time applications.
>
> Please refer to the General Reply, where we address this concern with a detailed inference latency table. As demonstrated, our model maintains superior performance across various architectures while introducing only a minimal impact on latency, making it suitable for real-time applications.

---

> ### Author Response · Authors · 2024-11-16
>
> > **W5**: The dictionary initialization process, particularly when using VLMs or LLMs, is not straightforward and may demand significant computational resources during setup. More details on optimizing or reducing this process would be beneficial.
>
>
> We would like to emphasize that our proposed method only requires a one-time initialization process. Compared to directly integrating large models like CLIP during training, our approach significantly minimizes the computational overhead for knowledge preparation. A single traversal of the entire dataset, combined with the use of LLM or VLM, requires just 1 hour on a single 3090 GPU. Additionally, we employ batch k-means and VAE (as shown in Table 5) to further accelerate the process efficiently.
>
> -----
>
> > **W6**: The module's efficiency in managing larger datasets or dynamic, continuously updated dictionaries remains unclear. Further analysis on handling diverse data scales could improve its adaptability.
>
> We have demonstrated the module's efficiency using MSCOCO, a large and diverse dataset. For smaller datasets, as shown in Table 9 (Appendix A.4), experiments on PASCAL-VOC confirm that our method consistently achieves superior results across different domains. Additionally, the classification task further highlights the module's adaptability to datasets of varying scales.

---

> ### Author Response · Authors · 2024-11-18
> **Supplementary Experiments on Q1**
>
> > **Q1**: Have you tried incorporating multple modality initialization and alignment for the dictionary?
>
> We extended our experiments to include additional combinations, such as (VM, VLM) and (VM, LLM), as per your query. Among these, the VM-VLM pair demonstrated the most notable performance. This finding aligns with *W3: Visualize*, where the analysis shows that VM and VLM exhibit a closer and more uniform distribution, resulting in their superior effectiveness when combined.
>
> |Combination | mAP(%) | mAP$_{0.5}$(%) |
> | :------: | :---: | :---: |
> |  VM, VLM, LLM* | 51.58 | 69.89 |
> |  VM, VLM* | 51.54 | 69.83 |
> |  VM, LLM* | 51.40 | 69.68 |
> | LLM, VLM  | 51.38 | 69.59 |
> |  VLM      | 51.75 | 70.12 |
> |  VM       | 51.37 | 69.42 |
> |  LLM      | 51.36 | 69.40 |
>
> *Note: Extended experiments

---

> > ### Comment · Reviewer_hT7W · 2024-11-25
> >
> > I have read other reviewers and the author response. I thanks for addressing my concerns.  I will wait for reviewers concern on computational analysis before finalizing my final rating

---

> > > ### Author Response · Authors · 2024-11-26
> > >
> > > Thank you for your reply. Below, we provide computational analysis, and including an additional GFLOPs comparison:
> > >
> > > | **Method**| **Knowledge Distillation** | **YOLO-World** | **RALF**       | **RD (Ours)**  |
> > > |------------|-------------|-------|---------|----------|
> > > | **Latency (ms)**| 4.00| 290.36| 19.43| 4.16|
> > > | **Training Cost**| 8 units × 4.5 days| 32 units × 2.8 days | 8 units × 3 days | 8 units × 2 days |
> > > | **GFLOPs**| 211.2| 371.4| 240.7| 109.8|
> > > | **mAP (%)**| 52.52| 51.00| 51.40| 53.36|
> > >
> > > The RD module requires only 3.3 GFLOPs, significantly less than other knowledge-based methods, while achieving superior performance and maintaining minimal computational overhead.
> > >
> > > The concern regarding the use of the `3x Faster R-CNN` schedule, raised by #itZp, we conduct additional experiments. The results are summarized as follows:
> > >
> > >
> > > |    | Baseline | Ours |
> > > |--------|----------|-------|
> > > | AP | 39.5| **40.9** |
> > >
> > > These consist demonstrate the efficiency of our method in more scenario.

---

### Official Review · Reviewer_DigY · 2024-11-03

**Soundness:** 3
**Presentation:** 4
**Contribution:** 3
**Rating:** 6
**Confidence:** 4

**Summary:**

The paper introduces the Retriever Dictionary (RD) module, a lightweight and efficient approach to integrating explicit knowledge into existing object detectors like YOLO, Faster R-CNN and DETR. RD outperforms other knowledge integration methods by introducing only 0.2M additional parameters while achieving superior performance. The module leverages pre-stored explicit knowledge within the Dictionary to enhance input features, demonstrating improvements across various YOLO-based tasks and broader computer vision applications like segmenation. The article also discusses ablation studies and visualizations that highlight the effectiveness of the RD module in retaining important information while eliminating unimportant details.

**Strengths:**

* Innovation: It presents a approach to integrate explicit knowledge into YOLO-based models by "efficiently" retrieving features from a dictionary built from insights across the entire dataset.
* Efficiency & Performance: RD achieves superior performance in object detection tasks, introducing only a minimal additional computational overhead~(with 0.2M additional parameters). And the paper shows it the lightest solution among compared methods.
* Visualization: The article includes visualizations that show RD's effectiveness in retaining important information while eliminating unimportant details.

**Weaknesses:**

1. While the paper effectively demonstrates the RD module's potential to enhance model accuracy with minimal computational increase, the inclusion of speed metrics would provide a more holistic view of the module's practicality and readiness for real-world deployment.
2. The knowledge integration methods compared in the article not only introduce external knowledge but also possess the capability for open-world detection. In contrast, the current work is primarily tailored for specific datasets. Although the paper demonstrates improved accuracy, the overall significance is diminished as a result.

**Questions:**

see Weaknesses

---

> ### Author Response · Authors · 2024-11-16
>
> > **Q1**: While the paper effectively demonstrates the RD module's potential to enhance model accuracy with minimal computational increase, the inclusion of speed metrics would provide a more holistic view of the module's practicality and readiness for real-world deployment.
>
> Thank you for your suggestion, we provide a table in General Reply, and we'll update the table in the paper. Generally, the table consistently shows our RD provide crucial information with an extremely low cost on both Params and Latency.
>
> > **Q2**: The knowledge integration methods compared in the article not only introduce external knowledge but also possess the capability for open-world detection. In contrast, the current work is primarily tailored for specific datasets. Although the paper demonstrates improved accuracy, the overall significance is diminished as a result.
>
> Our work focuses on leveraging underutilized dataset information and integrating multimodal knowledge into real-time detection models with minimal preparation, and enabling seamless integration into standard training. Additionally, RD is applicable across diverse architectures, demonstrating its broad applicability and general contribution to computer vision tasks.
>
> We appreciate your thoughtful feedback and agree that extending this approach to zero-shot or open-vocabulary tasks is a promising direction for future work.

---

> > ### Comment · Reviewer_DigY · 2024-11-25
> >
> > After considering the other reviews and the rebuttal, I maintain my score.

---

### Official Review · Reviewer_itZp · 2024-11-03

**Soundness:** 3
**Presentation:** 3
**Contribution:** 2
**Rating:** 6
**Confidence:** 3

**Summary:**

This paper proposes the Retriever Dictionary module, which improves the model’s resource utilization by incorporating knowledge from other parts of the dataset, extending beyond the local region or individual image. The Retriever module aggregates regional features to generate a query, while the Dictionary can be an unimodal or multimodal model, such as YOLOv7, CLIP, or GPT. Experiments with multiple detectors, including YOLOv7, YOLOv9, Faster R-CNN, and Deformable DETR, demonstrate performance improvements.

**Strengths:**

-  The idea of leveraging external knowledge from the dataset for object detection is promising, especially with foundational large language models (LLMs) and vision-language models (VLMs) enabling this approach.

- Applying dictionary learning to enhance object detection while maintaining a minimal increase in parameters is impressive.

**Weaknesses:**

- The primary weakness of this paper is the lack of computational analysis. The authors repeatedly emphasize "carefully balancing accuracy, model parameters, and latency when handling external information." However, latency is not addressed in the experiments. Although the added trainable parameters are minimal, the integration of a large foundational model significantly impacts the pipeline. It’s crucial to include runtime/latency and FLOPs metrics, particularly for YOLOv7 and YOLOv9 in Tables 1-3. This trade-off will help verify the effectiveness of the proposed approach.

- Employing different dictionaries demonstrates different behaviors. For instance, VLM improves YOLO models more than Faster R-CNN with LLM. This variation lacks justification, which seems essential to understand the behavior.


[1] Wasim, Syed Talal, et al. "VideoGrounding-DINO: Towards Open-Vocabulary Spatio-Temporal Video Grounding." Proceedings of the IEEE/CVF Conference on Computer Vision and Pattern Recognition. 2024.

[2] Liu, Shilong, et al. "Grounding dino: Marrying dino with grounded pre-training for open-set object detection." arXiv preprint arXiv:2303.05499 (2023).


Minor:
- The conclusion should be separated from the Experiment section.

**Questions:**

- The statement, "We demonstrate that incorporating external knowledge from models such as VLMs and LLMs can significantly enhance model performance," reflects an already established idea in object detection [1, 2]. How does this paper uniquely contribute to this area?

- The training strategy appears time-intensive. For example, Faster R-CNN is trained for 120 epochs, though a comparable baseline performance can typically be reached with 36 epochs (3x schedule). How do the authors justify this extended training duration?

---

> ### Author Response · Authors · 2024-11-16
>
> > **Q1**: The statement, "We demonstrate that incorporating external knowledge from models such as VLMs and LLMs can significantly enhance model performance," reflects an already established idea in object detection [1, 2]. How does this paper uniquely contribute to this area?
>
> While previous works [1, 2] often rely on 'online' models to provide knowledge, this approach significantly increases training and inference costs. For example, Grounding-DINO requires 80GB(A100) of memory for a single batch. In contrast, our RD module eliminates the need for an online model and enables fine-tuning of knowledge through dictionary learning. Notably, YOLOv7 with RD can be trained on devices with as little as 6GB of memory, showcasing the efficiency and practicality of our approach for real-time applications.
>
> > **Q2**: The training strategy appears time-intensive. For example, Faster R-CNN is trained for 120 epochs, though a comparable baseline performance can typically be reached with 36 epochs (3x schedule). How do the authors justify this extended training duration?
>
> As shown in Figure 10, during the training process our RD consistently outperformed the baseline. Moreover, we opted to train the model for a maximum of 120 epochs to minimize the impact of pretraining advantages and ensure a fair comparison across methods, notably, both experiments stopped early around 60 epochs.
>
> Furthermore, we trained Faster R-CNN from scratch using the same schedule. With RD, the performance closely matched that of pre-trained models, whereas models without RD showed significantly worse results. To maintain clarity and focus, we have excluded these specific experiments from the main comparison.
>
> > **W1**: The primary weakness of this paper is the lack of computational analysis. The authors repeatedly emphasize "carefully balancing accuracy, model parameters, and latency when handling external information." However, latency is not addressed in the experiments. Although the added trainable parameters are minimal, the integration of a large foundational model significantly impacts the pipeline. It’s crucial to include runtime/latency and FLOPs metrics, particularly for YOLOv7 and YOLOv9 in Tables 1-3. This trade-off will help verify the effectiveness of the proposed approach.
>
> Thank you for your suggestion. We have addressed this concern in the General Reply, which includes detailed latency and computational cost comparisons.
>
>
> > **W2**: Employing different dictionaries demonstrates different behaviors. For instance, VLM improves YOLO models more than Faster R-CNN with LLM. This variation lacks justification, which seems essential to understand the behavior.
>
> Faster R-CNN uses a ResNet backbone pre-trained on large-scale image datasets, already incorporating rich 'visual' knowledge. Thus, LLMs provide complementary linguistic insights, making a stronger impact. In contrast, YOLO models benefit more from VLMs due to without a large-scale dataset pretraining, making the alignment of visual and semantic spaces more impactful.

---

> > ### Comment · Reviewer_itZp · 2024-11-25
> >
> > The reviewer appreciates the authors' response, particularly the inclusion of computational analysis.
> >
> > However, my concern remains regarding the extended training duration. Specifically, Faster R-CNN trained with the [MMDetection 3x schedule](https://github.com/open-mmlab/mmdetection/blob/main/configs/faster_rcnn/faster-rcnn_r50_fpn_ms-3x_coco.py) is known to achieve comparable performance in significantly fewer epochs (36). While the authors state that RD consistently outperforms the baseline and justify the extended training to minimize pretraining advantages, this does not clarify why the dictionary-based module requires this prolonged schedule, especially when similar performance can be achieved with fewer epochs. Could the authors provide further justification or additional evidence to support the necessity of RD in this context?

---

> > > ### Author Response · Authors · 2024-11-26
> > >
> > > We have completed a pair of experiments using your specified configuration and employed `auto_scale_lr` to address GPU RAM inconsistencies. The results are as follows:
> > >
> > > | Method | AP   | AP$_{.5}$ | AP$_{.75}$ | AP$_s$ | AP$_m$ | AP$_l$ |
> > > |--------|-------|-----------|------------|--------|--------|--------|
> > > | Baseline | 39.5 | 57.6      | 44.3       | 24.6   | 44.4   | 50.2   |
> > > | Ours    | **40.9** | **61.7** | **44.5**   | **25.3** | **44.7** | **51.7** |
> > >
> > > Our method consistently demonstrates superior performance.
> > >
> > > Additionally, we noticed that the provided schedule includes specific optimizations, such as the `RepeatDataset` adjustments in `configs/common/ms_3x_coco.py` (lines 80, 95) and the `multiscale_mode` described in [this pull request](https://github.com/open-mmlab/mmdetection/pull/5179). These specialist designs help the performance.
> > >
> > > Our original intention was to evaluate RD under a standard baseline setting to ensure a clean comparison. Even when applied to the optimized schedule, our RD achieves consistent improvements under identical training conditions, confirming its effectiveness across settings.

---

> > > > ### Comment · Reviewer_itZp · 2024-11-26
> > > >
> > > > The reviewer appreciates the authors' efforts in validating the effectiveness of RD over the baseline with longer training schedules.
> > > >
> > > > After considering the authors' response, other reviews, and the overall contribution of the paper, I maintain my score and lean towards a weak acceptance of this paper.

---

> > ### Author Response · Authors · 2024-11-25
> >
> > Thank you for your feedback. Initially, we experimented with training from scratch (using ResNet pretraining) under different schedules. As shown below, the performance without RD was significantly lower:
> >
> > | 3x   |  Baseline   | Ours|
> > | -------- | ------|------ |
> > | AP | 29.9  |34.8|
> >
> > To ensure fair comparisons and robust results, we extended the maximum number of epochs in all subsequent experiments. At the 36th epoch under `FP16` training, the results were as follows:
> >
> > |    | Baseline    |Ours |
> > | -------- | --|--- |
> > |  AP| 38.30 |40.30 |
> >
> > Notably, the learning rate at later schedules drops to $10^{-5}$, meaning the 3x schedule already approaches the final results. This demonstrates the consistency and completeness of our training process.
> >
> > We can additionally run your suggested training schedule (3x, `FP32`, ResNet pertaining) if you want. While this will take approximately 15 hours, we will provide the results to supplement our comparison shortly.

---

### Official Review · Reviewer_BYfS · 2024-11-04

**Soundness:** 2
**Presentation:** 3
**Contribution:** 2
**Rating:** 6
**Confidence:** 4

**Summary:**

This paper aims to improve object detection with YOLO by introducing explicit knowledge in the form of a Retriever-Dictionary (RD) module. The dictionary is created using features from LLMs, VLMs, or VMs. The additional module helps the model retrieve relevant features from a dictionary that contains distilled information about the training dataset. The paper shows that such an addition leads to improved model performance.

**Strengths:**

The main strength of the paper is the increase in performance that is achieved by the proposed approach. The paper shows that the proposed approach leads to more than 3% performance improvements for object detection with YOLO. This result is quite good and demonstrates that the proposed approach might be useful. The paper also shows that the proposed approach can be used with other object detection models.

**Weaknesses:**

The major weaknesses of the paper are related to the lack of clarity about the motivation for the proposed approach and lack of clarity about several decisions. The authors should address the following in their response:

1. It's not clear, why is the dataset dictionary needed at all. Why doesn’t batch gradient descent already incorporate dataset information during training? The paper does not make it clear why/how is the dataset information not used during standard model training.

2. In lines 262-263, the paper mentions that the dimension of the dictionary features are the same dimension as YOLO middle layer dimension. Did the authors try incorporating dataset information at multiple locations in the network? It would help to see whether adding features at multiple levels (perhaps using dictionaries of different feature dimensions) improves performance.

3. Further, why add dictionary feature information at only the middle layer? What about any other layer?  Are there any draw-backs to using another layer?

4. The objective function written in section 3.3 (no equation number), needs to be explained better. It's not clear at all what is the objective trying to achieve and how does it achieve that. Such an explanation would help in understanding the motivation/approach more readily.


Edit: The authors have addressed most of my and other reviewers' questions satisfactorily in their answers and updates. Therefore, I am raising the rating to 6.

**Questions:**

Please see the weaknesses section.

---

> ### Author Response · Authors · 2024-11-16
>
> > **Q1**: It's not clear, why is the dataset dictionary needed at all. Why doesn’t batch gradient descent already incorporate dataset information during training? The paper does not make it clear why/how is the dataset information not used during standard model training.
>
> This is a great question. To clarify, we consider standard model training does utilize dataset information but uses weight inefficiently. The Retriever-Dictionary (RD) module explicitly captures dataset-level insights, whereas batch gradient descent only learns these insights implicitly by using whole model weights during training. By introducing this explicit and specialization mechanism, the RD module enables models to achieve performance that matches or exceeds that of much larger models without RD.
>
> For example, as shown in the Experiment, the RD module achieves next-level model performance—typically requiring a 100% increase in parameters—with only a 1% increase in parameters. This highlights its remarkable efficiency in leveraging dataset information.
>
> -----
>
> > **Q2**: In lines 262-263, the paper mentions that the dimensions of the dictionary features are the same dimension as the YOLO middle layer dimension. Did the authors try incorporating dataset information at multiple locations in the network? It would help to see whether adding features at multiple levels (perhaps using dictionaries of different feature dimensions) improves performance.
>
> We chose the middle layer because its feature space naturally aligns with the RD location, and offers a balance between generalization and specificity. Features from multiple layers might introduce alignment issues due to varying semantic meanings.
>
> ----
>
> > **Q3**: Further, why add dictionary feature information at only the middle layer? What about any other layer? Are there any drawbacks to using another layer?
>
>
> The RD module is placed at the location before entering the FPN, as shown in Appendix A.2, to provide a direct path to most subsequent blocks and layers. Placing it earlier would also reduce feature generalization due to proximity to the input and significantly increase model size due to exponential feature growth. Prior experiments showed a 0.5% mAP drop and an additional 1M parameters when the RD module was moved to a later layer. The current placement effectively balances performance and computational cost.
>
> ----
>
> > **Q4**: The objective function written in section 3.3 (no equation number), needs to be explained better. It's not clear at all what is the objective trying to achieve and how does it achieves that. Such an explanation would help in understanding the motivation/approach more readily.
>
>
> Thank you for pointing out this issue, we have updated the section for improved clarity.
> The objective function is designed based on both contrastive learning and knowledge distillation. Unlike traditional knowledge distillation, which trains on hard labels, our approach leverages contrastive learning to provide soft labels. This method aligns the distribution and relationships between the RD and the smaller retrieval dictionary, effectively avoiding overfit on hard labels. This alignment ensures robust knowledge transfer while maintaining flexibility for downstream tasks.

---

### Author Response · Authors · 2024-11-16
**General Reply**

We sincerely thank the reviewers for their insightful and encouraging feedback. We are delighted that the reviewers appreciate and acknowledge the RD module as a lightweight and flexible design that effectively integrates multimodal knowledge to enhance real-time object detection, demonstrating significant improvements across multiple models and tasks.

-----

One common concern raised was the potential extra latency introduced by the RD module. To address this, we have updated Table 1 with detailed latency measurements. These results report inference latency in milliseconds per batch, calculated with a batch size of 32 on an NVIDIA RTX 3090, evaluated across the entire MSCOCO validation set.

| **Latency (ms)** | **YOLOv7** | **YOLOv9** | **FasterRCNN** | **Deformable DETR** |
|-------------------|------------|------------|----------------|---------------------|
| **Without RD**    | 3.59       | 4.00       | 41.00          | 41.10              |
| **With RD**       | 3.70       | 4.16       | 41.02          | 41.28              |
| **Extra Latency**    | 0.11       | 0.16       | 0.02           | 0.18               |

The latency increase with RD is minimal—less than 3% for YOLOv7—highlighting its ability to provide critical knowledge with negligible computational cost.

We also compared RD against other knowledge-based methods in Table 3, demonstrating that our approach achieves superior training cost, latency, and performance:

| **Method**        | **Knowledge distillation** | **YOLO-World** | **RALF**      | **RD (ours)** |
|--------------------|--------------|----------------|---------------|---------------|
| **Latency (ms)**   | 4.00 | 290.36         | 19.43         | 4.16          |
| **Training Cost**       | 8 units × 4.5 days | 32 units × 2.8 days | 8 units × 3 days | 8 units × 2 days |
| **mAP(%)**       | 52.52 | 51.00 | 51.40 | 53.36 |

(*1 unit refer to 1 NVIDIA V100 GPU.)

These results confirm that RD not only achieves top performance with less training costs but also remains computationally efficient compared to alternative methods, showing its practicality for real-time applications.

---

### Meta-Review · Area_Chair_qhMz · 2024-12-19

**Metareview:**

This paper proposes a Retriever Dictionary (RD) module to introduce explicit knowledge of a dataset for enhancing object detection. Experiments with multiple detectors, including YOLOv7, YOLOv9, Faster R-CNN, and Deformable DETR, demonstrate performance improvements.

The main strengths are: 1) innovative ideas that leverage external knowledge from the dataset with LLMs and VLMs for object detection and 2) good performance.
The main weaknesses are: 1) lack of computational analysis, especially regarding latency, latency, and FLOPs metrics should be included, and 2) insufficient explanation and justification of how the model differs from RAG in network structure.

In the rebuttal, the authors provided computational analysis and more explanations, which well address the concerns raised by the reviewers.
The AC agrees with the reviewers and recommends accepting this paper, but the authors should consider the reviewers' suggestions to include necessary experimental comparisons.

**Additional Comments On Reviewer Discussion:**

All reviewers recognized the strengths of the idea of leveraging external knowledge from the dataset with LLMs and VLMs for object detection, and the issues raised by the reviewers are well addressed by the authors in the discussion stage.

---

### Decision · Program_Chairs · 2025-01-22

Accept (Poster)